# Learning Intrinsic Symbolic Rewards in Reinforcement Learning

## Abstract

Learning effective policies for sparse objectives is a key challenge in Deep Reinforcement Learning (RL). A common approach is to design task-related dense rewards to improve task learnability. While such rewards are easily interpreted, they rely on heuristics and domain expertise. Alternate approaches that train neural networks to discover dense surrogate rewards avoid heuristics, but are high-dimensional, black-box solutions offering little interpretability. In this paper, we present a method that discovers dense rewards in the form of low-dimensional symbolic trees - thus making them more tractable for analysis. The trees use simple functional operators to map an agent's observations to a scalar reward, which then supervises the policy gradient learning of a neural network policy. We test our method on continuous action spaces in Mujoco and discrete action spaces in Atari and Pygame environments. We show that the discovered dense rewards are an effective signal for an RL policy to solve the benchmark tasks. Notably, we significantly outperform a widely used, contemporary neural-network based reward-discovery algorithm in all environments considered.

## 1 Introduction

RL algorithms aim to learn a target task by maximizing the rewards provided by the underlying environment. Only in a few limited scenarios are the rewards provided by the environment dense and continuously supplied to the learning agent, e.g. a running score in Atari games (Mnih et al., 2015), or the distance between the robot arm and the object in a picking task (Lillicrap et al., 2015). In many real world scenarios, these dense extrinsic rewards are sparse or altogether absent.

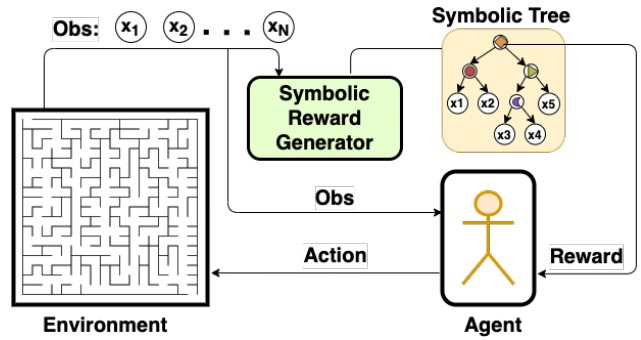

Figure 1: LISR: agents discover latent rewards as symbolic functions and use it to train using standard Deep RL methods

In these environments, it is common approach to hand-engineer a dense reward and combine with the sparse objective to construct a surrogate reward. While the additional density leads to faster convergence of a policy, creating a surrogate reward fundamentally changes the underlying Markov Decision Process (MDP) formulation central to many Deep RL solutions. Thus, the learned policy may differ significantly from the optimal policy (Rajeswaran et al., 2017; Ng et al., 1999). Moreover, the achieved task performance depends on the heuristics used to construct the dense reward, and the specific function used to mix the sparse and dense rewards.

Recent works (Pathak et al., 2017; Zheng et al., 2018; Du et al., 2019) have also explored training a neural network to generate dense local rewards automatically from data. While, these approaches have sometimes outperformed Deep RL algorithms that rely on hand-designed dense rewards, they have only been tested in a limited number of settings. Further, the resulting reward function estimators

are black-box neural networks with several thousand parameters - thus rendering them intractable to parse. A symbolic reward function lends itself better applications such as formal verification in AI and in ensuring fairness and removal of bias in the polices that are deployed.

In this paper, we present a method that discovers dense rewards in the form of low-dimensional symbolic trees rather than as high-dimensional neural networks. The trees use simple functional operators to map an agent's observations to a scalar reward, which then supervises the policy gradient learning of a neural network policy. We refer to our proposed method as Learned Intrinsic Symbolic Rewards (LISR). The high level concept of LISR is shown in Figure 1.

To summarize, our contributions in this paper are:

- We conceptualize intrinsic reward functions as low-dimensional, learned symbolic trees constructed entirely of arithmetic and logical operators. This makes the discovered reward functions relatively easier to parse compared to neural network based representations.
- We deploy gradient-free symbolic regression to discover reward functions. To the best of our knowledge, symbolic regression has not previously been used to estimate optimal reward functions for deep RL.

## 2 RELATED WORK

The LISR architecture relies on the following key elements:

- Symbolic Regression on a population of symbolic trees to learn intrinsic rewards
- Off-policy RL to train neural networks using the discovered rewards
- Evolutionary algorithms (EA) on a population of neural network policies search for an optimal policy

**Learning Intrinsic Rewards:** Some prior works (Liu et al., 2014; Kulkarni et al., 2016; Dilok-thanakul et al., 2019; Zheng et al., 2018) have used heuristically designed intrinsic rewards in RL settings leading to interesting formulations such as surprise-based metrics (Huang et al., 2019). In this work, we benchmark against Pathak et al. (2017) where a *Curiosity* metric was successfully used to outperform A3C on relatively complex environments like VizDoom and Super Mario Bros. LISR differs from these works in that the reward functions discovered are low-dimensional symbolic trees instead of high-dimensional neural networks. Further, unlike LISR, we are not aware other works that benchmark a single intrinsic reward approach on both discrete and continuous control tasks as well as single and multiagent settings.

**Symbolic Regression in DL** is a well known search technique in the space of symbolic functions. A few works have applied symbolic regression to estimate activation functions (Sahoo et al., 2018), value functions (Kubalík et al., 2019) and to directly learn interpretable RL policies in model based RL (Hein et al., 2018). To the best of our knowledge, symbolic regression has not previously been used to optimize for the reward function of an RL algorithm. For simplicity of design, we adopt a classic implementation where a population of symbolic trees undergo mutation and crossover to generate new trees.

**Evolutionary Algorithms** (EAs) are a class of gradient-free search algorithms (Fogel, 1995; Spears et al., 1993) where a population of possible solutions undergo mutate and crossover to discover novel solutions in every generation. Selection from this population involves a ranking operation based on a fitness function.

Recent works have successfully combined EA and Deep RL to accelerate learning. Evolved Policy Gradients (EPG) (Houthooft et al., 2018) utilized EA to evolve a differentiable loss function parame-terized as a convolutional neural network. CERL (Khadka et al., 2019) combined policy gradients (PG) and EA to find the champion policy based on a fitness score. Our work takes motivation from both. Like EPG, we also search in the space of loss functions - albeit in the form of low-dimensional symbolic trees. Like CERL, we allow EA and PG learners to share a replay buffer to accelerate exploration. However, unlike LISR, CERL relies on access to an environment-provided dense reward function for the PG learners.

## 3  LISR: LEARNING INTRINSIC SYMBOLIC REWARDS

The principal idea behind LISR is to discover symbolic reward functions that then guide the learning of a policy using standard policy gradient methods. A general flow of the algorithm is shown in Figure 2.

Two populations, comprising EA and SR learners respectively, are initialized. The EA population evolves using standard EA processes using a fitness function. In the SR population, each SR learner has a corresponding symbolic tree that maps state observations to a scalar reward. The nodes of the tree represent simple mathematical or logical operators sampled from a pre-defined dictionary of operators or *basis functions*. The complete list of basis functions that are utilized by the symbolic trees is described in Appendix B. The symbolic trees evolve using crossover and mutation based on a fitness function - leading to the discovery of novel reward functions. Figure 3 depicts these operations on symbolic trees.

Each SR learner uses its reward to update its weights via policy gradient (PG) methods. We adopt Soft Actor-Critic (Haarnoja et al., 2017) for the continuous control tasks and Maxmin DQN (Lan et al., 2020) for the discrete control tasks as the algorithms of choice since they are both state-of-the-art methods in those respective environments. In either case, the reward used to compute policy gradients is always an intrinsic, symbolic rewards and not any explicit dense reward provided by the environment.

The fitness function for any policy (SR or EA) is computed as the undiscounted sum of rewards received from the environment, which is given only at the completion of an episode. Thus, any dense reward provided by the environment is seen by any agent (SR or EA) only as a sparse, aggregated fitness function. At the end of each generation, all policies, EA and SR, are combined, ranked and a champion policy is selected.

The **shared replay buffer** is the principal mechanism enabling sharing of information across the EA and the SR learners in the population. Unlike, traditional EA where the data is discarded after calculating the fitness, LISR pools the experience for all learners (EA and SR) in the shared replay buffer - identical to standard off-policy deep reinforcement learning algorithms. All SR learners are then able to sample experiences from this collective buffer and use it to generate symbolic intrinsic rewards from

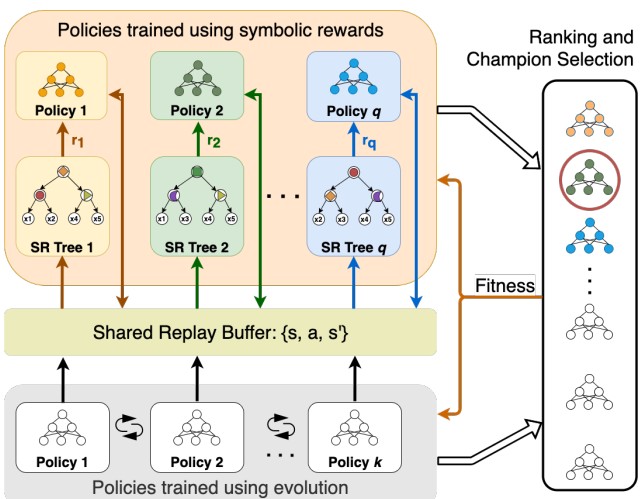

Figure 2: LISR: EA (bottom) and SR (top) learners share a common replay buffer. A set of symbolic trees sample observations from this buffer and map them into scalar rewards. SR learners also sample the same observations and the corresponding reward to train using policy gradients. The champion policy (circled) is selected by ranking all policies, EA and SR, based on a fitness function.

their respective symbolic trees and update the policy parameters parameters using gradient descent. This mechanism maximizes the information extracted from each individual experiences.

This architecture is motivated by CERL (Khadka et al., 2019) where a common replay buffer between evolutionary and policy gradient learners was shown to significantly accelerate learning. In our experiments, we vary the proportion of EA and SR policies in order to distil the incremental importance of each to the final performance. For completeness, the LISR is shown in Algorithm 1.

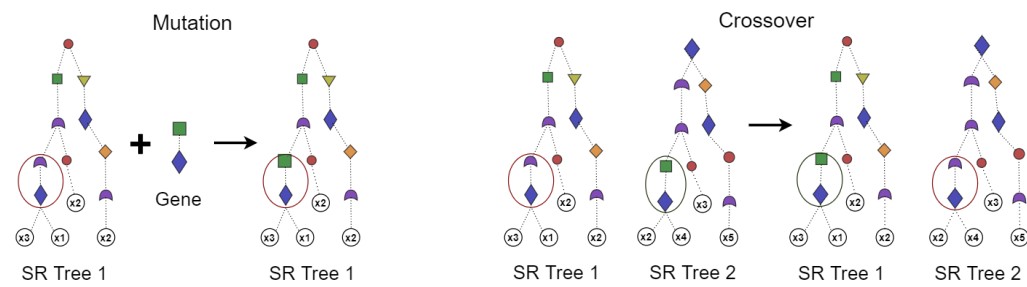

Figure 3: Evolution of symbolic trees. The colored polygons represent basic mathematical operators. For mutation (left), a random sub-tree is replaced using another random sub-tree (gene). For crossover (right), two parent trees swap sub-trees to form a child tree.

---

**Algorithm 1** LISR Algorithm

---

1: Initialize portfolio $\mathcal{P}$ with SR learners $\rightarrow$ (Algorithm 2)
2: Initialize a population of $k$ EA actors $pop_\pi$
3: Initialize an empty cyclic replay buffer $\mathcal{R}$
4: Define a random number generator $r() \in [0, 1)$
5: **for** generation = 1, $\infty$ **do**
6:     **for** actor $\pi \in pop_\pi$ **do**
7:         fitness, R = Evaluate($\pi$, R) $\rightarrow$ (Algorithm 3 in Appendix)
8:     Rank the population based on fitness scores
9:     Select the first $e$ actors $\pi \in pop_\pi$ as elites
10:     Select $(k-e)$ actors $\pi$ from $pop_\pi$, to form Set $S$ using tournament selection with replacement
11:     **while** $|S| < (k - e)$ **do**
12:         Use single-point crossover between a random $\pi \in e$ and $\pi \in S$ and append to $S$
13:     **for** Actor $\pi \in$ Set $S$ **do**
14:         **if** $r() < mut_{prob}$ **then**
15:             Mutate($\theta^\pi$) $\rightarrow$ (Algorithm 4 in Appendix)
16:     **for** Learner $L \in \mathcal{P}$ **do**
17:         Sample a random minibatch of T transitions $(s_i, a_i, s_{i+1})$ from $\mathcal{R}$
18:         Compute reward $\hat{r}_i = L_{\mathcal{ST}}(s_i, a_i, s_{i+1})$
19:         Compute $y_i = \hat{r}_i + \gamma \min_{j=1,2} L_{\mathcal{Q}'_j}(s_{i+1}, \tilde{a}|\theta^{L_{\mathcal{Q}'_j}})$
20:         Update $L_\mathcal{Q}$ by minimizing the loss: $\mathcal{L}_i = \frac{1}{T}\sum_i (y_i - L_{\mathcal{Q}_i}(s_i, a_i|\theta^{L_\mathcal{Q}}))^2$
21:         Update $L_\pi$ using the sampled policy actions
22:         Soft update target networks:
23:         $L_{\theta^{\pi'}} \Leftarrow \tau L_{\theta^\pi} + (1-\tau)L_{\theta^{\pi'}}$ and
24:         $L_{\theta^{\mathcal{Q}'}} \Leftarrow \tau L_{\theta^\mathcal{Q}} + (1-\tau)L_{\theta^{\mathcal{Q}'}}$
25:     **for** Learner $L \in \mathcal{P}$ **do**
26:         $score$, R = Evaluate($L_\pi$,R)
27:     Rank the learners $\mathcal{P}$ based on $scores$
28:     Select the first $j$ learners $L \in \mathcal{P}$ as elites
29:     Select $(m - j)$ symbolic trees $\mathcal{ST}$ from $\mathcal{P}_{\mathcal{ST}}$, to form Set $N$ using tournament selection.
30:     **while** $|N| < (m - j)$ **do**
31:         Use single-point crossover between a random $\mathcal{ST} \in j$ and $\mathcal{ST} \in N$ and append to $N$
32:         Use mutation between a random $\mathcal{ST} \in j$ and $\mathcal{ST} \in N$ and append to $N$

---

**Algorithm 2** Symbolic Reward Learner

---

1: **procedure** INITIALIZE
2:     Initialize actor $\pi$ and critic $\mathcal{Q}$ with weights $\theta^\pi$ and $\theta^\mathcal{Q}$, respectively.
3:     Initialize target actor $\pi'$ and critic $\mathcal{Q}'$ with weights $\theta^{\pi'}$ and $\theta^{\mathcal{Q}'}$, respectively.
4:     Initialize the symbolic tree $\mathcal{ST}$ for reward generation

## 4  EXPERIMENTS

Our main objective is to demonstrate that LISR can be applied to problems involving continuous and discrete action spaces. To this end, we evaluated LISR on Mujoco (Todorov et al., 2012) for continuous control tasks and on Pygame (Qingfeng, 2019) and OpenAI-Gym Atari games (Brockman et al., 2016) for discrete control tasks. We evaluated LISR's performance against three baselines: policies trained using a standard EA implementation, policies trained using only intrinsic symbolic rewards and *Curiosity* where agents learn on a combination of intrinsic rewards and environment-provided dense rewards.

For the continuous control tasks, we used **Soft Actor-Critic (SAC)** (Haarnoja et al., 2017) as our PG algorithm as it is the state-of-the-art on a number of benchmarks. SAC is an off-policy actor-critic method based on the maximum entropy RL framework (Ziebart, 2010). The goal of SAC is to learn an optimal policy while behaving as randomly as possible. This behavior encourages efficient exploration and robustness to noise and is achieved by maximizing the policy entropy and the reward.

For the discrete environments we adopt **Maxmin DQN** (Lan et al., 2020) which extends DQN (Mnih et al., 2015) to addresses the overestimation bias problem in Q-learning by using an ensemble of neural networks to estimate unbiased Q-values.

**Continuous control tasks:** We evaluated on four environments from the Mujoco benchmark - HalfCheetah, Ant, Hopper and Swimmer. We trained each environment with five random seeds for 150 million frames. We fixed the total population size (EA and SR learners) to 50 for all experiments. For LISR experiments, we kept the ratio between EA learners and SR learners equal to 0.5. For the *Curiosity* experiments, we integrated the *Intrinsic Curiosity Module (ICM)* to work with SAC. We performed a grid search for multiple learning rates for LISR, SR and *Curiosity* and report results corresponding to the best performing hyperparameters.

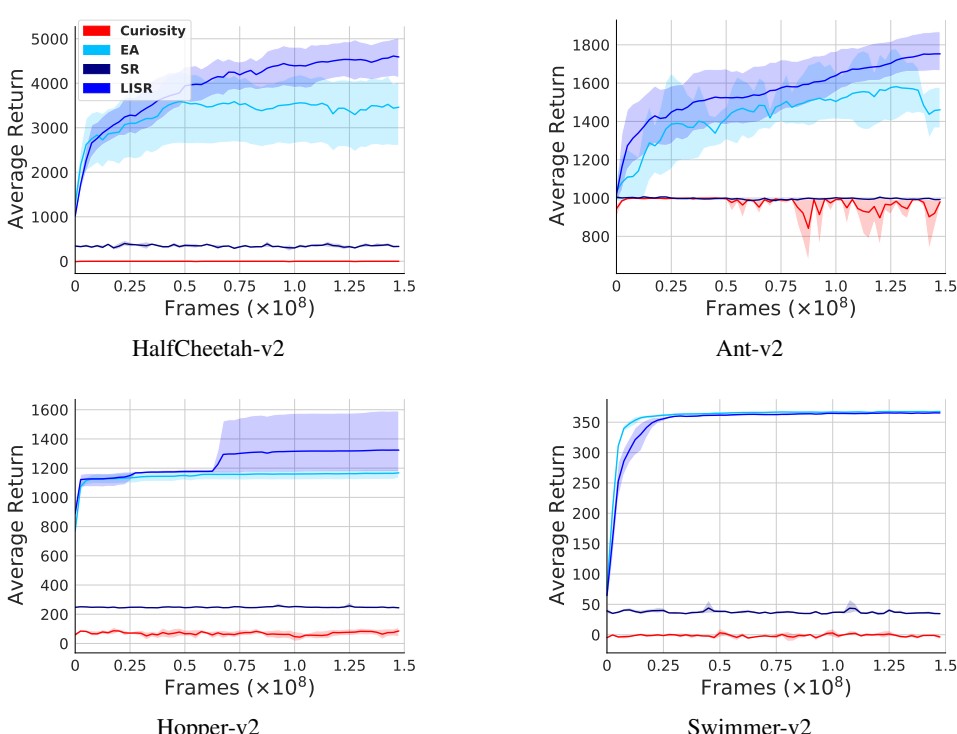

Figure 4: Results on continuous control tasks in Mujoco. LISR outperforms all baselines except on the low-dimensional problem in Swimmer. *Curiosity*, with access to explicit as well as implicit rewards, is unable to learn an effective policy on any environment. SR on its own, with access only to intrinsic rewards, is also unable to scale but slightly outperforms *Curiosity* .

Our results on the continuous control baselines are shown in Figure 4. We see that *Curiosity* fails to learn an effective policy on all environments - even though it has access to the dense rewards in addition to its own intrinsic rewards. SR on its own is also non-performant - however, it slightly outperforms *Curiosity* in 3 out of 4 environments. EA and LISR are both able to find effective policies. Notably, LISR outperforms EA substantially in 3 out of 4 environments. On Swimmer, EA slightly improves on sample efficiency - although both EA and LISR find the optimal solution quickly. This finding is consistent with Khadka et al. (2019) that also showed that EA outperformed reinforcement learning on the relatively low dimensional problem in Swimmer. Since the key difference between EA and LISR is the presence of SR learners, these results demonstrate the incremental importance of the discovered symbolic rewards in solving the tasks.

**Discrete control tasks:** We evaluated LISR on four different discrete environments: LunarLander and Amidar, two high dimensional environments from Atari games and PixelCopter and Catcher, two low dimensional environments from Pygames. We trained a multi-headed Maxmin DQN as our policy gradient learner and used the MeanVector regularizer (Sheikh & Bölöni, 2020) to ensure diversity in the Q-values. Similar to the baselines in the continuous control experiments, we evaluated the performance of LISR against the performance of only EA, only SR learners and *Curiosity* . We trained PixelCopter, LunarLander and Amidar for 50 million frames and Catcher for 30 million frames and show the results in Figure 5. We observe that similar to the continuous control tasks, LISR outperforms all the baselines except *Curiosity* in the Catcher environment where the performance of both LISR and *Curiosity* are similar. Notably, in the PixelCopter and Catcher environments, the SR learners alone were able to achieve the maximum performance - thus relying purely on discovered symbolic rewards. *Curiosity* significantly underperforms all baselines in all except the Catcher environment. The complete list of hyperparameters is shown in Appendix C.

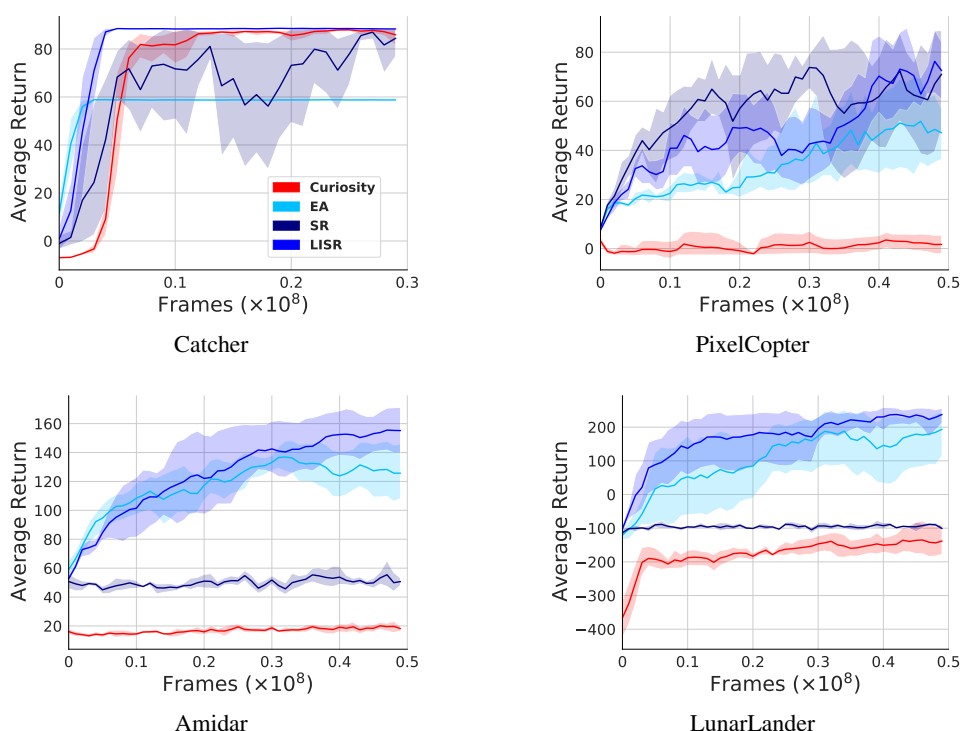

Figure 5: Results on discrete control tasks in Atari (top) and Pygames (bottom). LISR outperforms all baselines in all environments. *Curiosity* 's performance is overall significantly better on these tasks compared to continuous control. SR on its own, with no access to environment provided dense rewards, is able to completely solve the Atari environments. SR on its own also outperforms *Curiosity* on all but the Catcher environment.

**Multiplayer football**: We also applied LISR to Google Research Football (Kurach et al., 2020), a physics-based, multiplayer 3D environment with discrete action spaces where multiagent teams aim

to score goals and maximize their margin of victory. The environment provides a *Scoring* reward based on goals scored and a denser *Checkpoint* reward based on the distance of the ball to the goal.

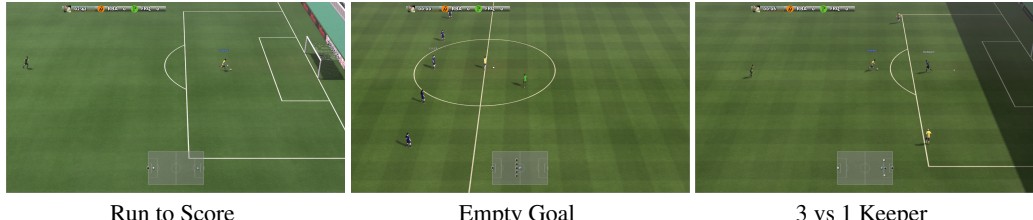

| Run to Score | Empty Goal | 3 vs 1 Keeper |

Figure 6: Google Research Football environments used in the experiments

We test LISR on 3 environment in the *Football Academy* set of environments - which describe specific game scenarios of varying difficulty. Specifically, we consider the following scenarios.

- *Run to Score*: Our player starts in the middle of the field with the ball, and needs to score against an empty goal. Five opponent players chase ours from behind.
- *Empty Goal*: Our player starts in the middle of the field with the ball, and needs to score against an empty goal.
- *3 vs 1 with Keeper*: Three of our players try to score from the edge of the box, one on each side, and the other at the center. Initially, the player at the center has the ball, and is facing the defender. There is an opponent keeper.

In the first two scenarios, we only control one player. In the last scenario, we consider variations where we control only one of our players and two of our players. In all cases, any player that we do not control utilizes the strategy of a built-in *AI bot*. The scenarios are shown in  Figure 6.

We benchmark LISR against EA and their published results with IMPALA (Espeholt et al., 2018) - a popular distributed RL framework that was shown to outperform other popular algorithms like PPO (Schulman et al., 2017) and variations of DQN (Horgan et al., 2018) on this benchmark. For LISR, we only use the aggregated sum of rewards in an episode as a sparse fitness function. IMPALA, on the other hand, utilizes a standard RL setup that exploits the dense rewards to learn a policy. Our goal was to investigate if LISR can be competitive with IMPALA with no access to the dense rewards.

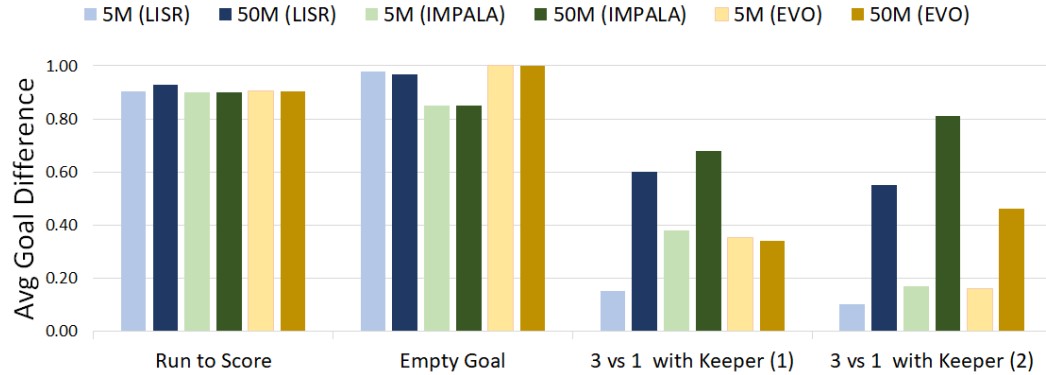

Figure 7: Experiments on Google Research Football environments. Numbers in parentheses indicate the number of players controlled by LISR

Figure 7 shows the performance on the four scenarios we evaluated. On the simpler environments involving an empty goal, all three algorithms were able to find performant solutions in less than 5M time steps. For the more difficult scenarios in involving 3 players vs 1, IMPALA does outperform LISR. However, LISR is able to find competitive strategies compared to IMPALA in both scenarios.

This is significant as it shows that even in relatively complex, non-stationary multiagent scenarios, LISR is able to discover intrinsic symbolic rewards and be competitive with well-established algorithms that exploit dense rewards.

**Discovered Rewards**: A key motivation to design LISR is the discovery of **symbolic** reward functions that are involve many fewer operations than a typical neural network based reward estimator. In all our experiments, we restricted the depth of the symbolic trees to 3 operational layers in order to impose these constraints.

Consider the PixelCopter environment shown in Figure 8. It provides 8 state variables which are: $s_0$: position; $s_1$: velocity; $s_2$: distance to floor; $s_3$: distance to ceiling; $s_4$: next block's x distance to player; $s_5$: next block's top y location and $s_6$: next block's bottom y location and $s_7$: agent's action.

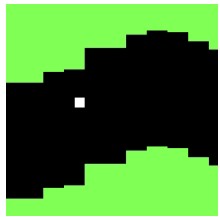

Figure 8: Pixel-Copter environment

Figure 9 shows an example of such a tree at the end of training. For better parsability, we unroll the tree into Python code. While we cannot claim that the particular reward function is interpretable, it is similar in structure to a classical symbolic rule and appears to rely on trigonometric transformations of positional variables. In this instance, it only utilizes 22 operations - thus making it relatively easier to analyze compared to neural network reward estimators. In contrast, *Curiosity* 's ICM module that generates an intrinsic reward is implemented as three neural networks with 5634 parameters.

```python
def get_intrinsic_reward(s_0, s_1, s_2, s_3, s_4, s_5, s_6, s_7):
    p_1 = tan(cos(s_4)); p_2 = cos(s_3); p_3 = pass_smaller(p_1, p_2)
    x_1 = multiply(-1, abs(subtract(s_7, p_3)))
    q_1 = multiply(-1, abs(subtract(1, s_4)))
    q_2 = max([s_2, 1, s_7, q_1, 0])
    q_3 = max([q_2, s_7, cos(0), multiply(s_0, s_6), multiply(s_5, subtract(s_6, 1))])
    y_1 = div_by_10(q_3)
    y_2 = square(s_7)
    y_3 = protected_div(1, div_by_100(s_0))
    x_2 = gate(y_1, y_2, y_3)
    z   = equal_to(x_2, x_1)
    reward = add(0, pass_smaller(div_by_10(s_7), z))
    return reward
```

Figure 9: An example of a discovered symbolic reward on PixelCopter. We unroll the corresponding symbolic tree into Python-like code that can be parsed and debugged. $\{s_i\}$ represent state observations.

## 5   CONCLUSION

In this paper, we presented LISR - a framework that combines ideas from symbolic, rule-based machine learning with modern gradient-based learning. We showed that it is possible to discover intrinsic rewards completely from observational data and train an RL policy. LISR outperformed other approaches that rely on neural network based reward estimators.

Our work is an effort to bridge the interpretability gap in Deep RL. While we cannot claim that the discovered reward functions are *interpretable*, they are relatively easier to parse - comprising of tens of symbolic operations compared to thousands of operations common in a neural network estimator. At the very least, this structure lends itself being more "human readable" compared to black box solutions. For example, as shown in our example tree, LISR required only 22 operations to compute a reward in PixelCopter - including simple trigonometric transforms on positional variables and one *if-then-else* gating condition. In a scenario where a policy is unstable, it could be feasible to trace the cause of instability to a subset of those operations. This kind of "limited explainability" could be important for safety-critical applications like autonomous driving scenarios. Future work will focus on building on the level of interpretability of the discovered functions.

One important drawback of LISR is the lack of sample-efficiency compared to established methods like SAC. This is somewhat expected as LISR operates with the key disadvantage of not having a pre-defined dense reward signal. The primary bottleneck involves the search for an optimal reward function. In this work, we implemented EA as the search mechanism. Future work will explore other alternatives like Monte-Carlo Tree Search (MCTS) as well as explore ways to turn off search when a reasonably good reward function has been discovered.

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

APPENDIX

## A    ALGORITHMS

---
**Algorithm 3** Function Evaluate
---
1: **procedure** EVALUATE($\pi$, R)
2:     $fitness = 0$
3:     Reset environment and get initial state $s_0$
4:     **while** env is not done **do**
5:         Select action $a_t = \pi(s_t|\theta^\pi)$
6:         Execute action $a_t$ and observe reward $r_t$ and new state $s_{t+1}$
7:         Append transition $(s_t, a_t, r_t, s_{t+1})$ to $R$
8:         $fitness \leftarrow fitness + r_t$ and $s = s_{t+1}$
9:     Return $fitness$, R
---

---
**Algorithm 4** Function Mutate
---
1: **procedure** MUTATE($\theta^\pi$)
2:     **for** Weight Matrix $\mathcal{M} \in \theta^\pi$ **do**
3:         **for** iteration = 1, $mut_{frac} * |\mathcal{M}|$ **do**
4:             Randomly sample indices $i$ and $j$ from $\mathcal{M}'s$ first and second axis, respectively
5:             **if** $r() < supermut_{prob}$ **then**
6:                 $\mathcal{M}[i, j] = \mathcal{M}[i, j] * \mathcal{N}(0, 100 * mut_{strength})$
7:             **else if** $r() < reset_{prob}$ **then**
8:                 $\mathcal{M}[i, j] = \mathcal{N}(0, 1)$
9:             **else**
10:                 $\mathcal{M}[i, j] = \mathcal{M}[i, j] * \mathcal{N}(0, mut_{strength})$
---

## B    SYMBOLIC TREE DETAILS

The complete list of operators used for symbolic tree generation is shown below.

```python
def add(left, right):
    return left + right

def subtract(left, right):
    return left - right

def multiply(left, right):
    return left*right

def cos(left):
    return np.cos(left)

def sin(left):
    return np.sin(left)

def tan(left):
    return np.tan(left)

def max(nums):
    return np.maxmimum(nums)

def min(nums):
    return np.minimum(nums)
```

```python
def pass_greater(left, right):
    if left > right: return left
    return right

def pass_smaller(left, right):
    if left < right: return left
    return right

def equal_to(left, right):
    return float(left == right)

def gate(left, right, condtion):
    if condtion <= 0:
        return left
    else:
        return right

def square(left):
    return left*left

def is_negative(left):
    if left < 0: return 1.0
    return 0.0

def div_by_100(left):
    return left/100.0

def div_by_10(left):
    return left/10.0

def protected_div(left, right):
    with np.errstate(divide='ignore',invalid='ignore'):
        x = np.divide(left, right)
        if isinstance(x, np.ndarray):
            x[np.isinf(x)] = 1
            x[np.isnan(x)] = 1
        elif np.isinf(x) or np.isnan(x):
            x = 1
    return x
```

## C    IMPLEMENTATION DETAILS

The complete list of hyperparameters used for LISR experiments are given below. For football experiments, we used the same hyperparameters that we used in discrete control tasks.

Table 1: Hyperparameters for LISR for continuous control tasks

| Hyperparameter | Value |
|---|---|
| Population Size $k$ | 50 |
| Target Weight $\tau$ | $1e^{-3}$ |
| Actor Learning Rate | $[1e^{-3}, 1e^{-4}, 3e^{-5}]$ |
| Critic Learning Rate | $[1e^{-3}, 1e^{-4}, 3e^{-5}]$ |
| Replay Buffer | $1e^6$ |
| Batch Size | $[256, 1024]$ |
| Exploration Steps | 5000 |
| Optimizer | Adam |
| Hidden Layer Size | 256 |
| Mutation Probability $mut_{prob}$ | 0.9 |
| Mutation Fraction $mut_{frac}$ | 0.1 |
| Mutation Strength $mut_{strength}$ | 0.1 |
| Super Mutation Probability $supermut_{prob}$ | 0.05 |
| Reset Mutation Probability $resetmut_{prob}$ | 0.05 |
| Number of elites $e$ | 7% |

Table 2: Hyperparameters for LISR for discrete control tasks

| Hyperparameter | Value |
|---|---|
| Population Size $k$ | 50 |
| Target Weight $\tau$ | $1e^{-3}$ |
| Actor Learning Rate | $[1e^{-3}, 1e^{-4}]$ |
| Maxmin DQN Heads | 2 |
| Regularization Weight | $1e^{-8}$ |
| Replay Buffer | $1e^6$ |
| Batch Size | $[64, 256]$ |
| Exploration Steps | 5000 |
| Optimizer | Adam |
| Hidden Layer Size | 256 |
| Mutation Probability $mut_{prob}$ | 0.9 |
| Mutation Fraction $mut_{frac}$ | 0.1 |
| Mutation Strength $mut_{strength}$ | 0.1 |
| Super Mutation Probability $supermut_{prob}$ | 0.05 |
| Reset Mutation Probability $resetmut_{prob}$ | 0.05 |
| Number of elites $e$ | 7% |

