# OpenReview forum: "Learning Intrinsic Symbolic Rewards in Reinforcement Learning"
_ICLR.cc/2021/Conference — Reject_

### Official Review · AnonReviewer3 · 2020-10-28
**Learning Intrinsic Symbolic Rewards in Reinforcement Learning**

**Rating:** 5
**Confidence:** 4

**Review:**

Summary of review:
Interesting work, but insufficient support of the main claim on interpretability.

Description:
This paper proposes a mechanism to learn reward functions that is based on a symbolic reward generator (in the form of a tree of pre-defined simple math operators).  The generated reward is used to optimize a policy, and an evolutionary approach is used over a set of these policies to select the final “champion” policy.  Observed rewards from the environment are used to rank the policies (rather than to directly optimize the policy).

The main claim of the paper is that this method is more interpretable, since presumable the learned reward function, in the form of this tree of simple operators, can be more easily interpreted than a general neural network (or other complex non-linear) approximator.

Strengths:
-	Paper is well written, easy to understand.  Good use of images to illustrate the method.
-	The proposed method is based on recent literature, and seems algorithmically sound.
-	There are several empirical results to characterize the performance, on diverse benchmarks and different baselines.
-	The authors are fair in characterizing their results.  They do not overclaim interpretability of the results presented (Fig.9, Sec. 4 & 5).

Weaknesses:
-	The motivation, of extracting interpretable rewards, is an interesting one.  Though the authors do not make a very convincing case for when this is absolutely necessary. The reality is that for most real-world tasks in RL (and I have worked on many!), the reward is specified by a human, and thus is interpretable from the start.  I can imagine some cases in imitation learning or inverse RL, where one has expert trajectories (e.g. driving a car in dense traffic) where it might be useful to infer an interpretable reward, but this is not the setting here.
-	The paper does not describe the formal setting.  It took me a while to figure out if/how observed rewards were used.
-	The authors do not provide a (time) complexity analysis of their algorithm, compared to others.
-	Some of the plots (Fig.4, 5) are hard to read, colours are too similar.  Fig.7 does not include error bars.
-	The empirical results on the symbolic reward being interpretable are not supported.  I tried to read the example in Fig.9, and it makes no sense.  Quantities with very different units are shoved into a function together, etc.
-	There are too many results on performance (which is not a main claim) vs interpretability (which is the main motivation).

Questions:
-	While I am not convinced by the interpretability of the method, the empirical results are still quite good. In general, I would expect to pay a performance cost for a more interpretable solution (that is usually the case in many other works on interpretable ML, due to using a simpler function.  Is it possible that here, the good performance is due to the symbolic reward estimator acting as a regularization mechanism?  Do you have any other explanation why the results are good?   Can you discuss in more detail the trade-offs between interpretability & performance?
-	I am not sure I understand exactly what each of the alternative methods considered in the empirical results are doing. Do you have a case that just uses the observed reward directly for policy gradient?   Which case would this be?

---

> ### Author Response · Authors · 2020-11-14
> **Response 1 of 1**
>
> Thank you for your feedback. We directly address your concerns here.
>
> **Weak interpretability compared to human defined rewards. Why is LISR needed?**
> You are correct that when a human can design an effective reward function, interpretable intrinsic rewards are not necessary for solving the underlying tasks. However, as RL becomes more and more prevalent in more domains with different tasks and requirements, we believe that it will soon become unscalable to have effective hand-designed reward functions for every task. As such, we are proposing a method that automatically generates reward functions where the designer has access to an intuitive episodic success metric. The generation of intrinsic rewards is motivated by any sparse rewards scenario. The case for interpretable intrinsic rewards is motivated by situations where a designer is motivated to understand the dense reward more thoroughly, such as safety critical applications or other situations where formal verification may be needed.
>
> **Description of the formal setting - if/how rewards are observed**
> During training time, we sample a batch of transitions from the replay buffer. We discard the observed reward and instead utilize the intrinsic reward generated from the symbolic reward tree. This intrinsic reward is then used to train the policy gradient algorithm.
>
> **Time complexity of the algorithm**
> Thank you for this suggestion. We did not track the time complexity formally. As any other population based RL method, our approach consumes negligible compute for policy updates compared to policy rollouts. In a distributed CPU cluster with roughly 48 nodes, we typically took around 2-3 days to train LISR. This is a significantly longer time compared to methods that have access to the explicit environment reward - but is comparable to intrinsic reward methods like Curiosity that need to discover effective rewards before a good policy can be trained.
>
> **Fig 4, 5 hard to read - colors are too similar. Example in Fig 9 makes no sense; quantities with different units are shoved into a function together, etc.**
> We will update the figures with better color schemes.
>
> Fig 9 shows an example of a discovered reward as Pythonized code. It is important to note that the reward generator has no notion of units - nor any notion of what physical quantities are represented. Thus, from the point of view of the reward generator, they are all unit-less quantities. This is no different from any other intrinsic reward method where the same observations would have been fed to a neural network - e.g., Curiosity.
>
> Our goal in this paper is not to attempt interpretability - but to discover significantly smaller reward estimators compared to neural networks. Further, your very observation that units of various quantities are disregarded by the reward generator indicates that a symbolic reward function can be “debugged” by a human expert. We think interpretability is a very open problem - but by enabling simple things like debuggability, we can bridge the gap to some extent.
>
> For future work, we intend to put more formal frameworks that specifically target interpretability. It is not inconceivable that such a framework would penalize the system for disregarding physical units.
>
> **Too many results on performance vs interpretability (which is the main motivation)**
> Interpretability is certainly an important motivation behind LISR. However, we want to emphasize that performance still remains a central priority.
>
> **Is the symbolic reward estimator acting as a regularization mechanism? Why are the results so good? Trade-off between interpretability and performance.**
> In essence, the estimator is acting as a platform to search for dense rewards that align with the sparser global (end-of-episode) reward. It can be seen as a regularizer in the sense that it selects for alignment with the global reward as it tries to optimize the fitness score.
>
> **Is there a case where the observed reward is used directly for policy gradient?**
> PG baselines like IMPALA do exactly this. We show our performance against IMPALA in Fig 7. Expectedly, we do not outperform IMPALA (since they use the observed reward whereas we discard the environment-provided reward and rely on discovered intrinsic rewards. However, we see that for simple scenarios (Run to Score and Empty Goal), we are equivalent to IMPALA in performance. The gap becomes larger with more complex environments involving multiple agents - e.g., we achieve a score of 0.6 compared to 0.8 for IMPALA in the 3-vs-1 scenario where two agents are trained. Future work will investigate ways to close this gap further.

---

### Official Review · AnonReviewer1 · 2020-10-28
**More details and analysis about the method are needed**

**Rating:** 4
**Confidence:** 4

**Review:**


The paper proposes LISR, a method for learning symbolic intrinsic reward functions from interaction using using symbolic regression via "symbolic trees". The approach is tested on a few continuous control Mujoco tasks, as well as a few discrete control gym environments and a few benchmarks based on the Google Research Football (GRF) environment.

Good things:

- The idea of using symbolic regression to discover intrinsic reward functions is extremely interesting, and a potentially good usage of symbolic regression towards enabling better interpretability of complex RL learning pipelines.
- Testing on multiple environments with wildly different tasks, control systems, and agent settings is excellent, and shows that the method is potentially applicable to a wide variety of settings.

Concerns:

1. The explanation of the method is somewhat poor, and almost entirely left to the (nonetheless good!) algorithm. The overall system integrates a lot of moving parts: a RL policies, multiple disjoint populations, a simple symbolic regression model, a shared experience replay buffer, etc. -- However, the manuscript is lacking of rationalisation about choices made when assembling this system is lacking (and effectively only the intuition behind the shared replay buffer is properly provided to the reader). This raises multiple questions, such as:
  a. Why were two different types of populations used?
  b. Why is the fitness evaluation mechanism shared among these two sets?
  c. Why was the fitness system used to rank policies, rather than reward models?
  d. Why was all of this not achievable simply through backpropagation?

My guess is that much of these details are contained in the CERL paper, however considering that the proposed pipeline largely seems to resemble it, it would probably be best to provide an overview of it.

2. It seems a little unfair to claim that LISR was tested on "multiagent" scenarios. GRF was mostly used as a single-agent environment, and the manuscript lacks in details about the setting with two agents. It would be good to add details about how the 2-agent setting was constructed (e.g. is there a centralised policy? Is there a joint action space? Is the reward function decomposed?), to understand exactly how LISR is operating in this setting.

3. Considering the significant amount of moving parts in the method, experimental evaluation section could use more ablations of the system and some more analysis. It would be good to understand for instance:
   a. how the system behaves with varying sizes of the populations;
   b. whether the reward trees look similar between multiple training runs / seeds;
   c. How the learnt trees differ between each environment setup...
... and so on.
This is particularly important for the narrative of the paper, since the method aims to improve interpretability of reward functions in RL agents for real-life tasks. It could also provide data to come up with suggestions on how to improve LISR to bridge the gap against other SOTA algorithms.


To conclude,  I currently cannot recommend acceptance, however I'd be willing to revise my score provided that at least some of the following improvements are made:
- Better description of the method, its assumptions, and how it specifically builds on / differs from CERL;
- More details about the output of the trained reward systems, and some analysis about why they produce what they produce (beyond empirical testing on multiple envs);
- More details about the multi-agent setup.

---

> ### Author Response · Authors · 2020-11-14
> **Response 2 of 2**
>
> **Similarity of reward trees between multiple seeds/runs/environments**
> Since our goal was primarily to achieve better performance on intrinsic reward problems via low-dimensional reward trees, we did not devote space to analyzing the properties of the learnt trees. Your suggestion on studying this is well taken and we will add a discussion on this in the appendix in the final manuscript.
>
> **How to bridge the gap with other SOTA algorithms?**
> Bridging the gap in performance against algorithms that are designed to operate on dense reward is not our primary goal. The value of LISR is in settings where meaningful dense rewards are **not** available in the environment. Most SOTA algorithms designed with dense rewards in mind fail to be effective in these scenarios - and often need to rely on heuristic methods of reward shaping. In this paper, we compare with a SOTA algorithm that operates under the constraints of intrinsic rewards (Curiosity) and demonstrate that we outperform it significantly on discrete and continuous action spaces.
>
> Importantly, we do so by constraining our discovered reward functions to be 2 orders of magnitude smaller than the neural network based reward estimators used in Curiosity.
>
> **Analysis of trained reward systems and why they produce what they produce**
> We provide an example of a Pythonized code (Fig 9) corresponding to a reward function that was discovered while training on PixelCopter. An analysis of the interpretability of the reward or the corresponding policy is outside the scope of this paper. We would like to re-iterate that our goal or claim is not centered around interpretability - rather around bridging the gap between complete black box neural network reward functions and hand-designed symbolic reward functions. One can think of this dimension reduction as a first step towards interpretability - and we do not claim any more than that in this paper.

---

> ### Author Response · Authors · 2020-11-14
> **Response 1 of 2**
>
> Thank you for your feedback. We directly address your concerns here.
>
> **Why were two different types of populations used?**
> We have two separate populations due to the nature of how the different populations are trained. In the evolutionary population, we search in the space of policy parameters using the episodic fitness value. In the LISR population we generate symbolic trees that serve as an intrinsic, dense reward for a regular RL agent (policy gradient in continuous action space, MaxminDQN in discrete action space) that trains in an off-policy fashion. Information within the population itself and across populations is shared across a shared replay buffer to bootstrap training for the LISR agents.
>
> Prior work such as CERL and MERL have demonstrated that such bi-level optimization involving evolutionary search and gradient based learning perform well for sparse reward problems. Since our episodic rewards are inherently sparse, we adopt this framework with two populations
>
>
>
>
> **Why is the fitness evaluation shared among those two sets?**
> The EA population needs a ranking function to evolve policies over time. SImilarly, the symbolic reward population needs a ranking function to evolve symbolic reward trees over time. We also have a single task objective. In such a setting, it is intuitive to utilize the task objective as a common fitness function for both populations since it provides a natural ranking metric that is aligned with the overall goal. This formulation also eliminates the need for defining proxy rewards or performing other types of reward shaping.
>
> **Why rank policies rather than reward models?**
> For each generation, our goal is to rank the reward models. Each reward model is associated with one policy - i.e., that policy uses the generated reward to update its weights. We postulate that the goodness of a reward model is intuitively captured by the fitness of the corresponding policy. Thus, by ranking the policies according to their fitness effectively ranks the corresponding reward model.
>
> **Why not use backpropagation?**
> In regular backpropagation in RL settings, one requires a dense reward function that provides useful information for every step in the environment. When that is not present, one ends up in a sparse reward setting which is a very weak learning signal on which to compute gradients. One common way is to define a heuristic dense reward based on human intuition and perform reward shaping to combine it with the task objective.
>
> Methods like Curiosity (a state of the art intrinsic reward estimator) avoid heuristic modeling by training a separate neural network to generate dense intrinsic rewards. This allows the policy gradients to be computed on dense rewards.  However, they are still fundamentally limited by the sparsity of the task objective when estimating the dense reward itself.
>
> In LISR, since we adopt evolutionary search to discover symbolic rewards, we do not need to compute any gradients. We directly utilize the episodic reward as a fitness function. We postulate that the dense rewards generated this way are less susceptible to the sparsity of the episodic reward. These dense rewards thus form a less noisy signal for the policy gradients to utilize.
>
> As we show in our experiments, this hypothesis is supported empirically as Curiosity underperforms LISR in various settings.
>
> **Details about how the 2-agent setting was constructed**
>
> -- The 2-agent setup was decentralized with each agent action entirely independently from the other agent. The agent do not share observations or action.
>
> -- We follow the formalism of centralized training and decentralized execution where the agents share replay buffer to train but act entirely independently during execution. They do not share the same observation or have any dependence/knowledge on other agent’s actions in computing their own.
>
> -- Reward is not decomposed or post-processed in any form manually. We use the sparse global reward as it is provided by the environment. Any dense rewards provided by the environment are discarded when writing into the replay buffer. The decomposition and reward shaping is all learned by the LISR algorithm.
>
> **Varying population size**
> Thank you for this suggestion. We did not add a full study on hyperparameter tuning as our goal was to demonstrate that a significantly more performant and computationally tractable solution can be obtained (compared to neural network based reward estimation) with minimal hyperparameter changes. The results in the paper required little tuning. We are open to adding a study on hyperparameters in the Appendix - but note that it would be peripheral to the primary take-away in the paper.

---

### Official Review · AnonReviewer2 · 2020-10-30
**It is not clear which models are used in symbolic regression**

**Rating:** 5
**Confidence:** 4

**Review:**

This paper presents a method to learn symbolic regression (tree) to make analysis of reward function more interpretable and tractable. Authors combine the benefits of interpretable SR learner and diversity driven evolutionary algorithms. Authors conduct the training of policy in the shared replay buffer for off-policy RL.

Strong points
- An interpretable way of policy learning using symbolic reward functions
- New interpretable reward as shown in Figure 9.

Weak points
- Symbolic reward functions are not clearly written (or not self-contained)
- It is hard to find algorithmic novelty in the paper
- Empirical evaluations are not extensive.
- It would be better to include examples of new discovery of interpretable reward function

The ideas in the presented paper looks reasonable. However, the symbolic reward function and the procedure to learn such symbolic representation is not clearly written. Also, it is hard to see that the proposed algorithm present new qualitative (interpretable) discovery of reward function.

---

> ### Author Response · Authors · 2020-11-14
> **Response 1 of 1**
>
> Thank you for your feedback. We directly address your concerns here.
>
> **Symbolic reward functions are not clearly written**
> While we acknowledge that the symbolic reward functions we end up are not the most intuitive to understand, we do publish an example of a function in the Appendix. The Appendix also contains our list of atomic function operators that we use to construct all symbolic function trees. Could you please clarify what you mean when you say that the functions are “not clearly written nor self-contained”?
>
> **Algorithmic novelty**
> We respectfully disagree with the notion that our work provides no algorithmic novelty. To the best of our knowledge, this is the first paper to apply automatically generated symbolic rewards as intrinsic rewards for an RL setting. As we mention in our overall response, intrinsic rewards are especially necessary in sparse reward settings. Similar work, such as Curiosity, used neural networks to provide intrinsic rewards, which are less interpretable and require more parameters. Moreover, our experiments show that LISR outperforms Curiosity on several RL tasks in discrete and continuous action spaces.
>
> **Evaluations are not extensive**
> We believe that our set of experiments covers a wide range of scenarios relevant to intrinsic reward generation across discrete and continuous action spaces. We adopt a state-of-the-art intrinsic reward method (Curiosity), as well as pure evolutionary search as our baselines. These cover the space of neural network generated rewards and gradient-free search methods. We would be happy to hear suggestions you have on other evaluations to run that are relevant to an intrinsic reward RL scenario.
>
> **Examples of discovery of interpretable reward functions**
> As mentioned in our overall remarks, we do not claim very easy and intuitive interpretation of our symbolic reward trees. We do, however, believe that this is a step in the direction of producing more interpretable intrinsic rewards in RL settings. Interpretability itself is a much bigger open problem in the RL and deep learning community, which we believe is outside the scope of this work. We do provide an example of how discovered rewards look like in Fig 9. We also publish all possible atomic functions that can be used to construct a reward function in the Appendix.

---

### Author Response · Authors · 2020-11-14
**Interpretability is the motivation - not the goal of this paper**

We thank all reviewers for their very helpful feedback.

One point we wish to make clear is around interpretability. Several comments pointed out that the discovered rewards are not interpretable. We completely agree.

Our goal in this paper is not to discover interpretable reward functions. Rather, we tackle a class of problems where meaningful, dense rewards are not available from the environment. In this scenario, to train an RL policy (using policy gradients, for example), prior works have taken two broad directions:
-- Define a heuristic dense reward based on knowledge of the domain and the environment. This is the most common approach.
-- Discover an intrinsic reward that is parameterized as a black-box neural network. This "reward generator" is thus another black-box neural network. Our baseline Curiosity is a state of the art method that takes this route.

In our work, we also discover intrinsic rewards. However, we constrain ourselves to discover rewards only in the form of symbolic functions. This takes the form of a typical heuristically defined reward but alleviates the need for any domain or environment knowledge. Further, we restrict our symbolic trees to be only 4 levels deep. The tree is allowed to select from a very simple set of operators (arithmetic, trigonometric, etc) so that the resultant tree is "small" and utilizes functions that are easy to parse.

Now, this is not interpretable by any means. As we show in our Fig 9, our discovered rewards utilized approximately 25 simple operations. This makes it significantly more **parsable** compared to the equivalent Curiosity solution that discovers an intrinsic reward using 5600 weights.

An interesting characteristic of the symbolic rewards is that they are amenable to human interaction. For example, a human expert might immediately notice redundant operations or operations involving incompatible physical quantities and make a correction. This can be done at any level of hierarchy - for one specific node or over a sub-tree. This kind of flexibility is difficult to achieve in a function parameterized by a neural network with thousands of weights.

We believe such an approach to reinforcement learning has not been studied before and is an important research angle that would pave the path to better interpretability. We hope this clarifies the point that while interpretability is a strong motivator for this paper, it is not the primary goal.

---

### Decision · Program_Chairs · 2021-01-07
**Final Decision**

**Decision:**

Reject

**Comment:**

This paper proposes an algorithm to learn symbolic intrinsic rewards via a symbolic function generator. The policy optimizes this reward function and an evolutionary algorithm selects between a set of such policies. The core idea is that learning with such a symbolic reward function is useful in sparse reward environments and also enables better interpretability.

${\bf Pros}$:
1. The learnt reward function has a relatively simple form and is therefore interpretable
2. The experimental section is quite extensive ranging from diverse tasks, control systems and agent systems. However there are some issues about showing clear need of the proposed method

${\bf Cons}$:
1. There was a consensus among reviewers that the paper does not make a strong case for the symbolic reward generator. In the rebuttal the authors argued that as RL scales to real world problems, it will become necessary to use such a method. I can understand how it would be useful in the context of inverse RL or imitation learning. However, as R3 points out, in the cases considered in this paper, the rewards are fairly intuitive and explainable. The paper might become stronger by directly tackling problems with such constraints.
2. There is confusion about the details and scope in the current version of the paper. The paper would become stronger by incorporating all the feedback received during the review period.